# Furosemide Promotes Inflammatory Activation and Myocardial Fibrosis in Swine with Tachycardia-Induced Heart Failure

**DOI:** 10.3390/ijms26136088

**Published:** 2025-06-25

**Authors:** Nisha Plavelil, Robert Goldstein, Michael G. Klein, Luke Michaelson, Mark C. Haigney, Maureen N. Hood

**Affiliations:** 1Department of Radiology and Bioengineering, F. Edward Hébert School of Medicine, Uniformed Services University of the Health Sciences, Bethesda, MD 20814, USA; nisha14biochem@gmail.com; 2Department of Medicine, F. Edward Hébert School of Medicine, Uniformed Services University of the Health Sciences, Bethesda, MD 20814, USAmichael.klein@usuhs.edu (M.G.K.);; 3Department of Anesthesiology, F. Edward Hébert School of Medicine, Uniformed Services University of the Health Sciences, Bethesda, MD 20814, USA

**Keywords:** furosemide, diuretic, tachycardia, heart failure, ECM remodeling

## Abstract

Loop diuretics like furosemide are commonly used in heart failure (HF) treatment, but their effects on disease progression are still unclear. Furosemide treatment accelerates HF deterioration in a swine model, but the mechanism of acceleration is poorly understood. We hypothesized that furosemide activates inflammatory signaling in the failing left ventricular (LV) myocardium, leading to adverse remodeling of the extracellular matrix (ECM). A total of 14 Yorkshire pigs underwent permanent transvenous pacemaker implantation and were paced at 200 beats per minute; 9 non-instrumented pigs provided controls. Seven paced animals received normal saline, and seven received furosemide at a dose of 1 mg/kg intramuscularly. Weekly echocardiograms were performed. Furosemide-treated animals reached the HF endpoint a mean of 3.2 days sooner than saline-treated controls (mean 28.9 ± 3.8 SEM for furosemide and 32.1 ± 2.5 SEM for saline). The inflammatory signaling protein transforming growth factor-beta (TGF-β) and its downstream proteins were significantly (*p* ≤ 0.05) elevated in the LV after furosemide treatment. The regulatory factors in cell proliferation, mitogen-activated protein kinase signaling pathway proteins, and matrix metalloproteinases were elevated in the furosemide-treated animals (*p* ≤ 0.05). Our data showed that furosemide treatment increased ECM remodeling and myocardial fibrosis, reflecting increased TGF-β signaling factors, supporting prior results showing worsened HF.

## 1. Introduction

Unlike other heart failure (HF) treatments, the impact of loop diuretics has not been examined in a large randomized clinical trial. Despite this, the use of diuretics to reduce volume overload is considered a keystone in the treatment of decompensated HF patients. Diuretics help with reducing volume overload and congestive symptoms. Loop diuretics, such as furosemide, increase the loss of water through urine by inhibiting the luminal sodium–potassium–chloride channels (NKCCs) in the loop of Henle [1,2]. Loop diuretics are recommended for the rapid treatment of congestive symptoms [3] and are prescribed to 90% of hospitalized heart failure patients [4] However, the net effect of loop diuretics on the progression of human HF remains uncertain. In retrospective analyses of the Studies of Left Ventricular Dysfunction (SOLVD) and Digoxin (DIG) studies, the use of loop diuretics was associated with increased HF hospitalization and HF mortality [5,6].

Fibrosis is a consequence of HF and develops in the interstitial cardiac tissues called the extracellular matrix (ECM) via protein deposition [7]. ECM depositions are induced by intense release of fibrogenic mediators that are associated with inflammatory cytokines, chemokines, reactive oxygen species, the renin–angiotensin–aldosterone system (RAAS), and growth factors such as transforming growth factor-beta (TGF-β). TGF-β signaling is a primary cardiac fibrogenic mediator. TGF-β1 binds to the TGF-β1 receptors, activates Smad2/3 cascade, and induces the transformation of myofibroblast to fibroblasts [8]. In addition, TFG-β suppresses the catabolism of ECM by regulating the expressions of tissue inhibitors of metalloproteinases (TIMPs) and plasminogen activator factors [9,10]. This combination leads to the overproduction of ECM, which develops into cardiac fibrosis [11,12]. It has been previously demonstrated that TGF-β deficient transgenic mice showed significantly reduced ECM deposition and, thereby, less cardiac fibrosis [13]. These data support the pivotal role of endogenous TGF-β1 signaling in the progression of ECM deposition, cardiac fibrosis, and eventually increased HF.

From animals in a randomized study performed in a porcine model of HF by our group, it was found that furosemide accelerated the progression of systolic dysfunction and development of severe HF compared with saline-treated controls. In a previous study by our group, furosemide use increased activation of the RAAS, including aldosterone [14]. Excess aldosterone in HF has been associated with increased cardiac fibrosis [15], HF progression, and sudden death. Thus, we hypothesize that furosemide stimulates the pro-fibrotic pathways of the ECM and thus accelerates pathologic remodeling in HF.

## 2. Results

This study was designed to elucidate the effects of furosemide treatment on molecular and structural changes in a tachycardia-induced heart failure pig model. For the western blot specific pigs in this study, tachypaced pigs receiving furosemide (n = 10) reached the HF end point on average 3.2 days earlier (mean 28.9 ± 3.8 SE for furosemide and 32.1 ± 2.5 SE for saline), days (10%) earlier (*p* = 0.087) than those receiving saline (placebo) (n = 7), concordant with an earlier report by our group [14]. Despite a shorter pacing duration, the mean fractional shortening of the pigs was comparable at 13.0 ± 4.9 in HF furosemide and 12.76 ± 3.7 in HF saline groups (*p* = 0.84). At sacrifice, the left ventricle manifested significantly greater biochemical and histological evidence of fibrosis, as well as increased markers of inflammation.

### 2.1. Upregulated Mitogen-Activated Protein Kinase (MAPK) Signaling Proteins in Furosemide Administration

The mitogen-activated protein kinase (MAPK) signaling pathway plays an important role in the regulation of cell differentiation, cell proliferation, and apoptosis. There are three main families of MAPK: extracellular signal-regulated kinases (ERKs), c-Jun Nh2-terminal kinases (JNKs), and p38 and stress-activated protein kinases (SAPKs). Our data from left ventricular cardiac tissue showed significantly elevated expression in the furosemide-treated animals for all three families of the MAPKs as compared to saline and control animals (*p ≤* 0.05) (Figure 1A,B). Another protein related to the mitogen-activated extracellular signal-regulated kinase (MEK)-ERK pathway family is the rat sarcoma protein (Ras). Ras signaling is involved in cell proliferation as a part of the rat sarcoma protein (Ras)-Raf-MEK-ERK signaling pathway [16]. We found that the level of Ras elevated significantly in our furosemide-treated animals as compared to saline and control animals (*p* ≤ 0.05) (Figure 1C).

### 2.2. Elevated TGF β and Its Downstream Signaling Seen in Furosemide Treatment

TGF-β1 and its downstream proteins were found to be elevated in both tachycardia-induced HF animals. This elevation was consistently greater in the group receiving furosemide: The relative expression of TGF-β1 and TGF-β receptor 1 were significantly elevated in the furosemide-treatment HF animals when compared with the control and HF animals treated with saline (Figure 2A–C) (*p* ≤ 0.05). An increased plasma TGF-β1 level was also observed in the furosemide-treated animals compared to saline-treated and control animals β1 (Figure 2D) (*p* ≤ 0.05). Once activated, the TGF β receptor initiates the downstream signaling molecules Smad2 and Smad3. The levels of Smad2 and Smad3 were significantly increased in the furosemide-treated animals as compared to saline-treated and control animals (Figure 2E–G) (*p* ≤ 0.05).

### 2.3. Growth Factor Biomarkers in the ECM Remodeling

Galectin-3, a mammalian carbohydrate-binding lectin involved in inflammatory regulation and associated with pathologic remodeling in HF, was significantly elevated in the furosemide-treated animals as compared to saline-treated animals and controls (Figure 3A,B) (*p* ≤ 0.05). Additionally, the plasminogen activator inhibitor type 1 (PAI-1), a mediator of fibrotic development, was also significantly increased for furosemide animals compared to saline and control animals (Figure 3A,C) (*p* ≤ 0.05). During the cardiac remodeling, connective tissue growth factor (CTGF) is significantly induced in cardiac myocytes. We have analyzed the levels of growth differentiation factor-15 (GDF-15) and connective tissue growth factor (CTGF) in the left ventricular tissue, and the results showed that both the biomarkers were significantly higher in the furosemide-treated animals as compared to saline-treated animals as well as controls (Figure 3E,F) (*p* ≤ 0.05) in the myocardial tissue. In addition to the tissue level of the biomarkers, we have analyzed plasma levels of endothelin-1 (ET-1) and CTGF. The results showed an increased level of plasma ET-1 and CTGF in both the HF animals, but a significantly higher level of these biomarkers was observed in the furosemide-treated animals (Figure 3D,H) (*p ≤* 0.05). This is an important connection in that the plasma elevation is also reflected in the tissues. ET-1 is found to upregulate TGF-β1 and CTGF in processes that contribute to fibrotic remodeling [17], which is consistent with our findings.

### 2.4. Level of Matrix Metalloproteinases During ECM Remodeling

Proteins associated with fibrotic growth were found to be significantly increased in animals given furosemide. HF biomarkers were assessed from the left ventricle after tachycardic pacing are depicted in Figure 3. The matrix metalloproteinases are major regulators of cardiac remodeling in the extracellular matrix. Our results found that MMP-14 and TIMP-1 to be significantly elevated in left ventricular tissue of the furosemide HF animals as compared to both saline-treated and control animals (Figure 4A–C) (*p* ≤ 0.05). Along with this, the plasma level of MMP-13 and TIMP-2 were significantly elevated in the furosemide-treated animals compared with control and saline-treated animals (Figure 4D,E, *p* ≤ 0.05).

### 2.5. Histology of Collagen Deposition During ECM Remodeling

Fluorescent immunohistochemistry of collagen I, III, and VI, as well as Masson trichrome histology, were assessed in this study. The structural collagens I, II, and VI were found to be significantly elevated in the HF animals treated with furosemide as compared with saline-treated and control animals in our semi-quantitative analysis (Figure 5A–F) (*p* ≤ 0.05).

Histological analysis of the left ventricular tissue stained with Masson’s trichrome explained and confirmed the derangement of the ventricular tissue. Visually, the percentage of blue (fibrotic material) compared to myocytes for the furosemide group looks clearly higher compared to the saline and control groups (Figure 6). The mean percentage of blue fibrotic material compared to muscle tissue was Control = 1.22% ± SEM 0.25, Saline HF = 2.21 ± SEM 1.18, and Furosemide 12.51 ± SEM 8.4. A two-sided test comparison (Figure 6) also showed a strong significance value between the controls and furosemide-treated animals (*p* ≤ 0.05) but not between saline- and furosemide-treated animals. We think that the small, uneven group size may have contributed to the lack of clear differences in the groups.

## 3. Discussion

In this study, we assessed the impact of a commonly prescribed loop diuretic, furosemide, on inflammation and matrix remodeling in a large animal model of HF. In support of our hypothesis, our data indicated that furosemide significantly elevates inflammatory and stress-regulating proteins in heart tissue and plasma.

Although furosemide-treated HF patients experience immediate relief from their congestive symptoms, furosemide is known to activate the RAAS in patients and animals in heart failure [13,14]. Other investigators demonstrated significant increases in collagen deposition from an infusion of aldosterone within 6 weeks [18], suggesting rapid enhancement of pro-fibrotic processes may occur due to RAAS activation, such as when furosemide is used to treat HF. A study by Veeraveedu et al. [18] also found that although furosemide was a potent diuretic, it did not improve survival and was associated with increased TGF-β1 and collagen levels in the myocardium [19]. The TORNADO HF study compared furosemide to torasemide, with the furosemide group performing much worse in a 3-month period of time [20]. Furthermore, a study by Yamazaki et al. (2012), to evaluate diuretic use after myocardial infarction, used furosemide as a comparison to tolvaptan [21]. They found furosemide did not improve LV function and increased macrophage development and increased B-type natriuretic peptide (BNP), both of which signal pathologic activity [21] and may relate to the activation of the RAAS [22,23].

Key biomarkers in the MAPK and TGF-β1 pathways were elevated in the LV myocardium during HF, particularly in the presence of furosemide (Figure 1 and Figure 2). LV tissue analysis of galectin-3 expression showed a significant increase in both the HF animal groups compared to control pigs, plus significantly elevated in the HF furosemide group. Galectin-3 has prognostic value in HF, which is expressed at the early phase of HF pathology [24,25]. The elevation of galectin-3 in cardiac tissues in our study is consistent with a compensatory response of the myocardium attempting to counteract the effects of the inflammatory and fibrotic damage [26]. Plasminogen activator inhibitor-1 (PAI-1), a prothrombotic protein and an important regulator in the initial phase of cardiovascular diseases, was also found to be significantly elevated in both HF groups compared to controls, with a significant increase for furosemide-treated HF animals relative to saline-treated HF animals. Since angiotensin-II can upregulate the expression of PAI-1 [27], this is significant, as aldosterone is upregulated in tachycardia swine treated with furosemide [14].

Biomarkers relating to an increase in growth may be an early indicator that change is occurring and could help lead to early detection of adverse remodeling. Growth differentiation factor-15 (GDF-15) is an emerging predictor associated with heart failure and metabolic diseases, which can regulate LV remodeling in the heart [28,29]. The HF pig tissue showed a significantly increased expression of GDF-15 compared to control groups supporting the association of GDF-15 upregulation with tachycardia-induced HF.

Biomarkers related to stress and inflammation that can result from tachycardia (p38 MAPKs, ERKs, JNKs, and MAP kinase or ERK kinase-1 (MEK-1)) were significantly elevated in the furosemide animals as compared to saline-treated and control animals, supporting the evidence that furosemide can stimulate the MAPK signaling pathway [30]. The ERK 1/2 proteins in this pathway are signaling proteins that play a role in cell proliferation, which were also significantly elevated in the furosemide animals, suggesting that furosemide is stimulating pathological remodeling beyond that of the tachycardia.

Inflammatory cytokines are involved in the regulation of cardiac fibrosis, and TGF-β1 is a key pro-fibrotic cytokine, which is markedly elevated in cardiac remodeling [31,32,33,34,35]. Our results also revealed TGF-β1 signaling was elevated in the plasma, consistent with work by Almendral et al. (2010), reporting that elevated serum levels of TGF-β1 were linked to cardiomyopathy [36]. TGF-β1 ligands are well known to play a role with Smad proteins in the signaling of the extracellular milieu [37]. Much research has been devoted to understanding the induction of TGF-β signaling in HF; however, relatively little is known about the effects of furosemide on TGF-β signaling. Our results clearly show that the TGF-β1 signaling proteins are elevated in furosemide-treated HF animals, as compared to both saline-treated HF animals and controls. This may represent an important mechanism for the detrimental effect occurring from the use of furosemide on the inflammatory response induced by tachycardia. Our study was consistent with prior studies demonstrating the upregulation of TGF-β receptor 1 and Smads 2/3, leading to fibrosis [11,35,36,37,38].

The ECM is a highly adaptive, dynamically active scaffold that is influenced by a wide array of factors, such as mechanical stress, neurohormonal activation, inflammation, and oxidative stress [39,40,41]. Fibroblasts are the cells that maintain the structure of the ECM, which are maintained by a number of cytokines, enzymes, and peptides, with matrix metalloproteinases (MMPs) and tissue inhibitor metalloproteinases (TIMPs) playing major roles [42]. The activity and expression of MMPs are particularly elevated in the failing heart [43,44,45,46]. Our results found increases in both MMP-14 and TIMP-1, which are known to contribute to the remodeling process, but their exact mechanisms are not well understood but have been associated with HF [47,48].

Our results from immunohistochemistry and histology demonstrate increased inflammatory markers and collagen deposition in furosemide-treated, HF-induced animals, similar to increases in fibrosis from angiotensin II infusions by Li et al. [49], lending to our earlier assertion that RAAS activation was occurring in our model. Since both heart failure groups had a physical change in the myocardium that was determined by echocardiogram to have reached a fractional shortening of 16% or less, it is expected that some sort of change in the ECM of the animals in both groups was expected. Biochemical and immunohistochemical studies showed furosemide-increased factors leading to myocardial deterioration and fibrosis and play a significant role in the cardiac remodeling, which is evident in our model.

## 4. Materials and Methods

### 4.1. Animals

A total of 17 Yorkshire pigs (8 male and 9 female) (*Sus scrofa domesticus*; 7 to 10 weeks of age) underwent permanent transvenous pacemaker implantation and were ventricularly paced at 200 beats per minute to induce heart failure. They were randomized to daily injections of furosemide versus saline (placebo), as described previously [14]. A total of 7 (2 male and 5 female) non-instrumented pigs were studied as non-HF controls. After the initiation of pacing, active-treatment animals received furosemide (1 mg/kg intramuscularly), and placebo-treated animals received saline intramuscularly (placebo) every day. These animals were used for all the Western blots, ELISAs, and immunohistochemistry. A subset of these animals based on the tissue available were used for histology. The dose of furosemide was comparable to that given in previous studies using dogs [50]. Animals were housed 2 per cage, given enrichment toys, had ad libitum access to water, were not restricted with respect to salt, and were weighed every 5 days. Animals were checked daily for health issues by veterinary staff. All animals were housed and cared for the same manner in order to reduce confounders. Animals were sacrificed within 24 h of developing systolic dysfunction, as defined by a fractional shortening of 16% or less by echocardiogram [14]. After pentobarbital anesthesia, blood sampling, and lethal injection of potassium chloride, hearts were immediately removed, and left ventricular free wall tissue samples rapidly obtained. Blood and tissue were then stored at −80 °C. The protocol was reviewed and approved by the Institutional Animal Care and Utilization Committee and conforms with the guidelines endorsed in the “Position of the American Heart Association on Research Animal Use,” adopted 11 November 1984, by the American Heart Association [51] and the “Guide for the Care and Use of Laboratory Animals” [52].

### 4.2. Sample Preparation and Protein Analysis by Western Blot

Previously frozen (−80 °C) left ventricular tissue samples were obtained, ground and homogenized in NP-40 buffer (Invitrogen™(ThermoFisher Scientific, Waltham, MA, USA) NP40 Cell Lysis Buffer #FNN0021)) with an added protease inhibitor cocktail (Thermo Scientific, Waltham, MA, USA) Halt Protease Inhibitor Cocktail (100×) #78429), and centrifuged using 3000× *g* at 4 °C. Blood samples were collected, and serum was separated using 3000× *g* centrifugation at 4 °C. After the removal of the tissue debris by centrifugation, the concentration of the protein in the supernatant was measured using the bicinchoninic acid assay (BCA) Protein Assay Kit #23227 (ThermoFisher Scientific, Waltham, MA, USA). For Western blot (WB), protein samples (20 µg) were resolved by electrophoresis using 4–12% SDS–polyacrylamide gels (Invitrogen #NP0335, (ThermoFisher Scientific, Waltham, MA, USA)) under denaturing and reducing conditions and blotted to nitrocellulose membranes (Bio-Rad #1620215 (Bio-Rad, Hercules, CA, USA)). The membranes were blocked with blocking buffer (LI-COR #927-60001 (LI-COR Biosciences, Lincoln, NE, USA)) and then subjected to immunoblot analysis using standard methods [53]. The primary antibodies used for the immunoblots are listed in all the primary antibodies (Table 1) and were diluted in Intercept^®^ T20 (TBS) Antibody Diluent (LI-COR #927-65001 (LI-COR Biosciences, Lincoln, NE, USA)). The blotted membrane was kept at 4 °C overnight and washed 3 times with PBS-Tween 20. Then the membranes were treated with IRDye secondary antibodies (1:20,000) (LI-COR (LI-COR Biosciences, Lincoln, NE, USA), IRDye^®^ 680RD Donkey anti-Mouse IgG (H + L) (LI-COR #926-68072 (LI-COR Biosciences, Lincoln, NE, USA), (IRDye^®^ 680RD Donkey anti-Rabbit IgG (H + L) (Li-COR #926-68073 (LI-COR Biosciences, Lincoln, NE, USA),), diluted in Intercept^®^ (LI-COR Biosciences, Lincoln, NE, USA), T20 (TBS) antibody diluent for 2 h, and then washed 3 times with PBS-Tween 20. Protein fluorescent bands were digitalized immediately before signal saturation using an Odyssey image system (LI-COR Biosciences, Lincoln, NE, USA) and analyzed with Odyssey^®^ CLx Imaging System (Image Studio™, V 5.2, (LI-COR Biosciences, Lincoln, NE, USA)). The data were quantified as fluorescent intensity (arbitrary units). Each experiment was repeated at least 3 times to confirm reproducibility.

### 4.3. Enzyme-Linked Immunosorbent Assay (ELISA)

The protein levels of TGF-β1 (#RK03371), endothelin-1 (#RK03316), connective tissue growth factor (#RK03311), MMP-13 (#RK03352), and TIMP-1 (#RK03373) were quantitatively measured by ELISA kit (ABclonal, (Abcam, Inc. Waltham, MA, USA) according to manufacturer’s instructions. Briefly, 100 μL standard/sample diluent was placed in blank wells and followed by 100 μL of standard and sample in adjacent wells and then covered with the adhesive strip provided. Wells were incubated for 2 h at 37 °C, followed by an aspiration/wash and then incubated for 1 h at 37 °C with biotin-conjugated antibody. The wells were then washed, with 100 μL of streptavidin-horseradish peroxidase (HRP) diluent added, and incubated for 30 min at 37 °C. After a second wash and added 3,3′,5,5′-Tetramethylbenzidine (TMB) substrate (100 μL/well), wells were incubated for 15–20 min at 37 °C, followed by addition of stop solution (50 μL/well). The optical density of each well was measured within 5 min, using a microplate reader (Azure AC3000, Dublin, CA, USA) set to 450 nm.

### 4.4. Histological Preparation

Previously frozen transmural left ventricular sections of heart were kept in 10% formalin overnight (furosemide-treated samples = 4, saline-treated = 5, control = 9). The histology has a smaller number of samples as these were the only samples still available from the original set of animal tissue samples. The sections were treated with xylene–ethanol and then soaked in paraffin for 60 min before embedding into a paraffin mold. Tissue sections were cut at 5 μm thickness using a microtome (Leica RM2235, Biosystems, Deer Park, IL, USA). These sections were deparaffinized by xylene–ethanol treatment. Deparaffinized sections were stained using Masson trichrome reagents (Sigma-Aldrich # HT15-1KT (Sigma-Aldrich, Burlington, MA, USA)). Random images of the LV tissue were acquired on a Keyence BZ-X series phase contrast microscope (Keyence Corporation of America, Itasca, IL, USA). All images were acquired at 60× with oil immersion and white balance. The images were analyzed by calculating the percentage of fibrotic (blue pixels) versus muscle (red pixels) in HSV color-space for each image within the three groups, Control, HF Furosemide, and HF Saline, using a custom analysis routine written in IDL software V8.5.1 (L3Harris Geospatial—an NV5 Global Company, Broomfield, CO, USA). Each animal had 6–10 random pictures acquired, which were then averaged to give a mean percentage of fibrotic to muscle per animal. Since the data failed Levene’s test of homoscedasticity and the groups are small of uneven size and uneven numbers of pictures, a Kruskal–Wallis test was performed to compare groups using IBM SPSS Statistics (SPSS, Chicago, IL, USA, V 28.0.0) to assess for any trends, as this part of the study is underpowered.

### 4.5. Fluorescent Immunohistochemistry (ICH) from Frozen Tissue

Frozen LV sections were mounted with Optimal Cutting Temperature compound (OCT), and the specimens were kept at −80 °C for at least 12 h before sectioning. Mounted samples (10 um thick) were sectioned with a cryostat (Leica CM3050, Leica Biosystems, Buffalo Grove, IL, USA). The immunohistochemical analyses of the left ventricle of swine heart were carried out as previously described [54] with the following minor modifications. The tissue sections were washed with PBS and performed antigen retrieval by using 5 mM HEPES buffer with 1 mM ethylenediaminetetraacetic acid (EDTA) and 0.05% Triton X-100 at pH 8 and incubated at −80 °C for 10 min and blocked with 5% normal goat serum for 1 h. Then the slides were incubated with primary antibodies (Table 1, supplement) diluted 1:300 overnight at 4 °C. Slides were washed 3 times with PBS and incubated with Alexa Fluor-conjugated secondary antibodies, diluted 1:300 (AF 488, Jackson Laboratories, West Grove, PA, USA) for 1 h at room temperature in dark. Then slides were washed with PBS, and the sections were mounted with 4′,6-diamidino-2-phenylindole (DAPI)-fluoromount-G mounting medium (Southern Biotech, Birmingham, AL, USA. Cat #0100-20) and covered with cover slips. Fluorescence was visualized and imaged using the Leica DM-6 B fluorescence microscope (Leica Microsystems GmbH, Wetzlar, Germany) using the imaging software Leica Application Suite, X (LAS-X, Leica Microsystems GmbH, Wetzlar, Germany). Five animals in each of the three groups (Control, HF Furosemide, and HF Saline) had LV tissue processed for this study. The immuno-stained LV sections were imaged using a tiling program to acquire over 300 images in a group. This section of tiled images was then used to select five individual tiles (one from each corner and one from the middle of the tiles) to analyze for the intensity measurements. All the images were taken with the same exposure settings. Fluorescence was measured using the Leica DM6B microscope. The fluorescence data analysis was performed using Image Processing and Analysis Software in Java (ImageJ), V 1.53, (ImageJ, National Institutes of Health, Bethesda, MD, USA).

### 4.6. Statistical Analysis

Results for the Western blots and ELISAs are expressed as mean ± SEM. Statistical analysis was performed using Microsoft Office Excel (Office 2019, Redmond, WA, USA) and IBM SPSS Statistics (SPSS, Chicago, IL, USA, V 28.0.0). For the comparisons of the proteins, one-way ANOVA was used to compare between groups, with Tukey honestly significant difference (HSD) correction and descriptive statistics. For the histology comparisons, Kruskal–Wallis test was performed, and significance was set at *p* ≤ 0.05.

## 5. Limitations of the Study 

In this study, furosemide was initiated prior to the onset of systolic dysfunction, which represents a departure from usual clinical practice in humans [14]. In addition, we did not collect daily weights on the animals or collect information on hydration of the animals. Since diuretics can trigger a RAAS response due to the water/salt loss in the body as our group discussed in a prior publication [14], we need to further study if the addition of a neurohumoral blockade could counter the fibrotic effects. Our study was also limited by sample size, especially for the histology assessment of the fibrotic changes in the ECM. However, our findings in this limited study stress the need to further study the effects of furosemide on heart failure. 

## 6. Conclusions

This study provides novel evidence elucidating the involvement of furosemide in the accelerated progression of porcine HF and supports the findings of an earlier study by our group. Furosemide treatment in our tachycardia model increased the TGF-β and MAPK signaling cascades in swine, leading to the overproduction and deposition of various ECM proteins. Since swine have similar structural and physiological responses to humans, they make an excellent model to test therapies compared to other animal models, excluding non-human primates. Because of this similarity, we think that furosemide may exert a similarly adverse effect on human HF. However, we did not check hydration status in the animals; thus, we need to explore to see if the use of medications such as neurohumoral blockades to reduce RAAS activation can counter the fibrotic cascade. We hope that this study brings awareness that furosemide may increase mediators of LV inflammation and fibrosis and thus improve protective strategies when employing diuretics is prudent in HF management.

## Figures and Tables

**Figure 1 ijms-26-06088-f001:**
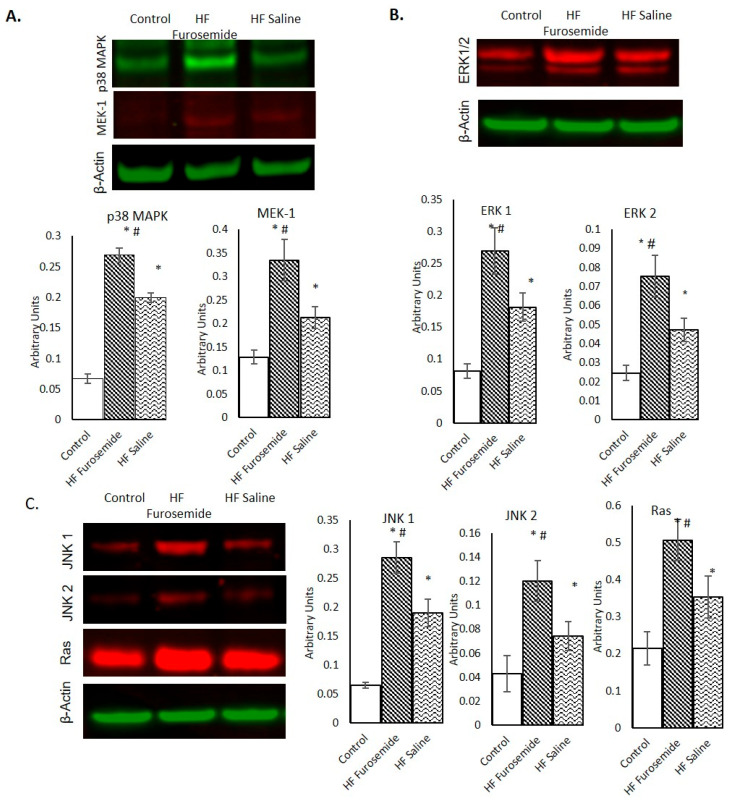
MAPK signaling molecules in the progression of heart failure. Expression in the furosemide-treated animals for proteins in the MAPK family shows an elevation in all the proteins. (**A**) Western blot analyses and densitometric quantitation of p38 MAPK and MEK-1 images from the infrared scanner. (**B**) Western blot analyses and densitometric quantitation of ERK1/2. (**C**) Western blot analyses and densitometric quantitation of JNK1, JNK2, and Ras in the LV homogenate of Normal, HF furosemide, and HF saline swine. The graphs of individual proteins demonstrate the relative intensity mean values, shown as mean ± SEM. The HF furosemide group for all proteins was found to be significantly different from both the control and HF saline groups, *p* ≤ 0.05 (n = 7). The saline HF means were also significantly different from the control animals (* = statistically different from control group, # = statistically different between furosemide HF and saline HF group).

**Figure 2 ijms-26-06088-f002:**
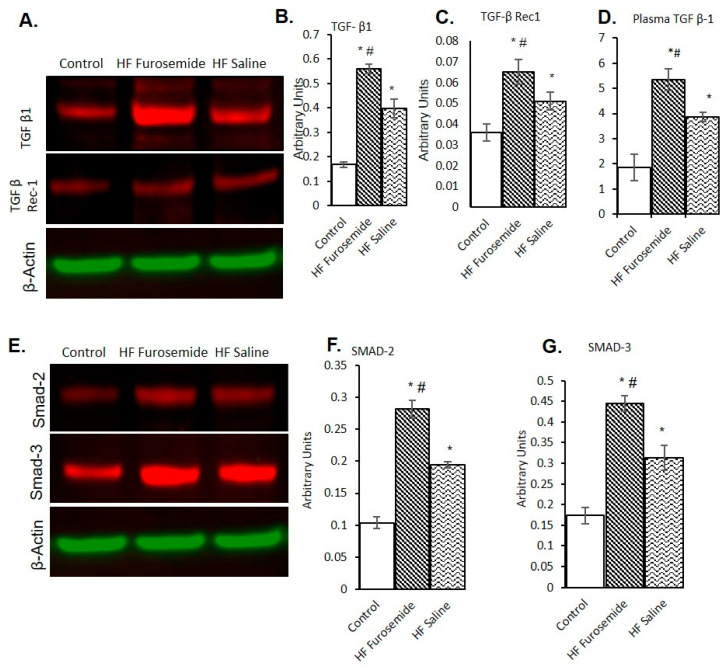
Proteins associated with the TGF-β1 pathway and inflammatory response. Western blots of a variety of proteins relating to heart failure from the tachycardia-induced pacing are depicted in (**A**). Western blot expression in furosemide-treated animals found elevated TGF-β1 and TGF- β receptor1 endogenously (**B**,**C**) and plasma level of TGF-β1 proteins to be elevated shown in (**D**), and the data shown as mean ± SEM, *p* ≤ 0.05. Similarly, a Western blot data for Smad-2 and Smad-3 are shown in (**E**–**G**), which are significantly increased in furosemide-treated animals compared with saline-treated and control animals, and the data shown as mean ± SEM, *p* ≤ 0.05, (* = statistically different from control group, # = statistically different between furosemide HF and saline HF group).

**Figure 3 ijms-26-06088-f003:**
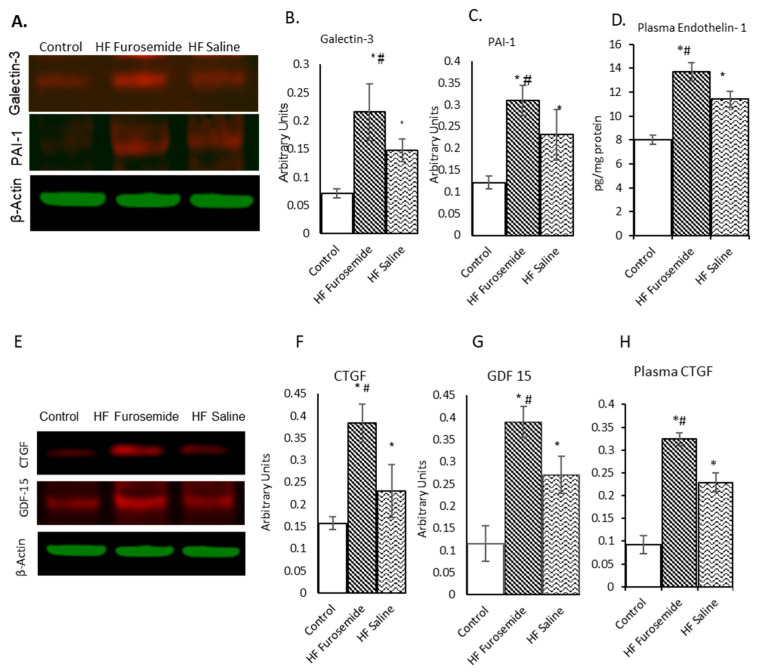
Biomarkers related to growth factor regulation in ECM remodeling. Western blot results for several growth factors involved in the regulation of growth on the extracellular matrix. Endogenous expression of galectin-3 (**A**,**B**), PAI-1 (**A**,**C**), CTGF (**E**,**F**) and GDF-15 (**E**,**G**), are significantly elevated in furosemide-treated animals compared to control and saline-treated HF animals (data shown as mean ± SEM, *p* ≤ 0.05). ELISAs were performed on plasma from a limited subset of the animals (control = 4, HF furosemide = 6, and HF saline = 8). Endothelin-1 has been associated with possibly inducing pro-inflammatory mechanisms in the heart. Endothelin-1 in plasma was found to be significantly elevated in the HF animals as compared to controls ((**D)**, data shown as mean ± SEM, *p* ≤ 0.05), but not significantly different between HF groups. The level of CTGF found to be significantly elevated in the left ventricular tissue is also elevated in the plasma (**H**) for the heart failure animals compared to control animals, plus furosemide HF-treated animals were significantly higher than the saline-treated HF animals (data shown as mean ± SEM, *p* ≤ 0.05). (* = statistically different from control group, # = statistically different between furosemide HF and saline HF group).

**Figure 4 ijms-26-06088-f004:**
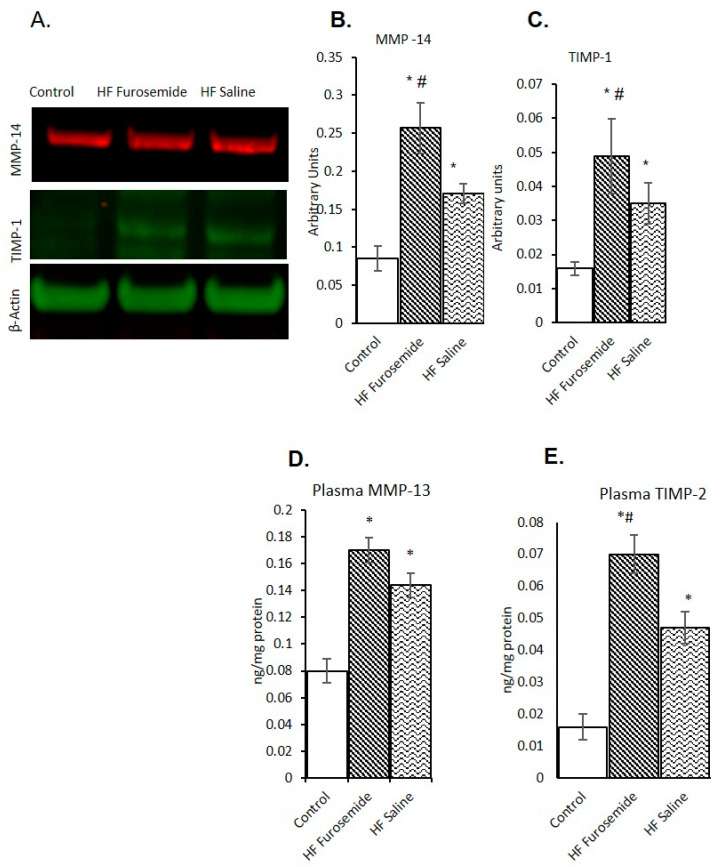
Level of matrix metalloproteins in tachycardia-induced heart failure. Western blot results for several proteins involved in the regulation of growth on the extracellular matrix. MMP-14 and TIMP-1 are competitors in the remodeling of the heart. MMP-14, which is activated in response to inflammation and injury, is significantly elevated in the furosemide animals as compared to both controls and saline HF animals (**A**–**C**, data shown as mean ± SEM, *p* ≤ 0.05). The saline HF animals are also significantly elevated as compared to the control animals, *p* ≤ 0.05. TIMP-1 is an inhibitor of the MMPs; it is also involved in the remodeling process and is elevated when there is tissue damage. The plasma levels of MMP-13 and TIMP-2 results show significantly elevated levels of MMP-13 and TIMP-2 levels in the furosemide-treated animals compared with control. There is no significant elevation of MMP-13 is seen in furosemide treatment when compared with saline treatment, but the TIMP-2 level is significantly increased in furosemide-treated when compared with saline administration (**D**,**E**, data shown as mean ± SEM, *p* ≤ 0.05). (* = statistically different from control group, # = statistically different between furosemide HF and saline HF group).

**Figure 5 ijms-26-06088-f005:**
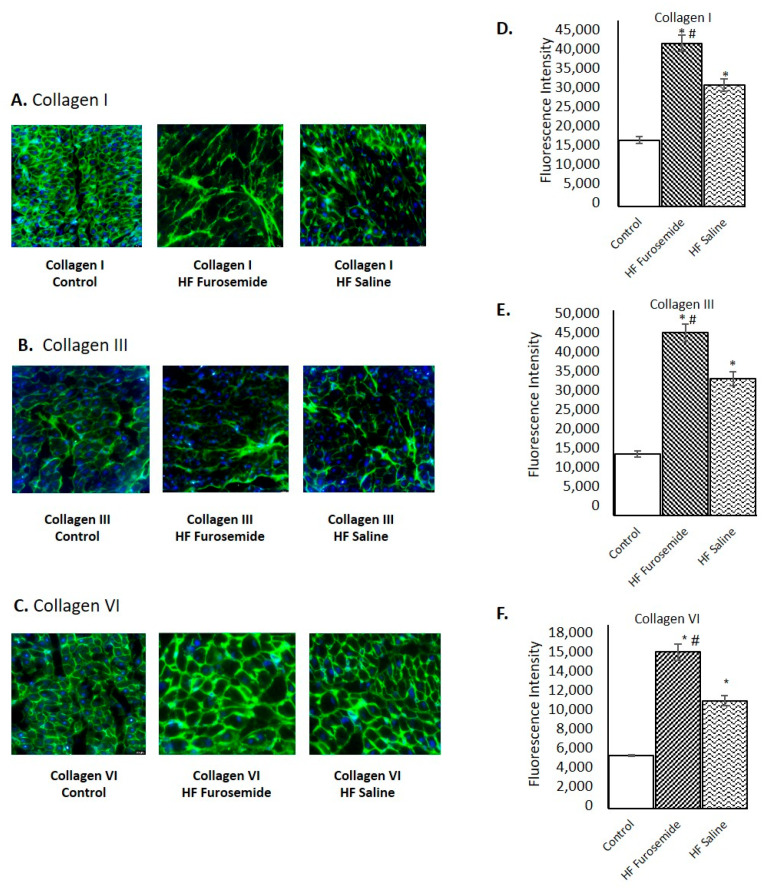
Immunohistochemistry of collagens I, II, and VI and tissue histology. Collagens I, II, and VI were assessed with immunohistochemistry on 5 µm slices obtained from the frozen blocks of left ventricular tissue (**A**–**C**). The bar graphs (**D**–**F**) show the relative fluorescent intensities of the collagens, presented as mean ± SEM, *p ≤* 0.05 (n = 5). All of the collagens show significantly elevated in the furosemide animals as compared to both controls and saline HF animals. Also, the saline HF animals are significantly elevated, as compared to the control animals, *p ≤* 0.05. (* = statistically different from control group, # = statistically different between furosemide HF and saline HF group).

**Figure 6 ijms-26-06088-f006:**
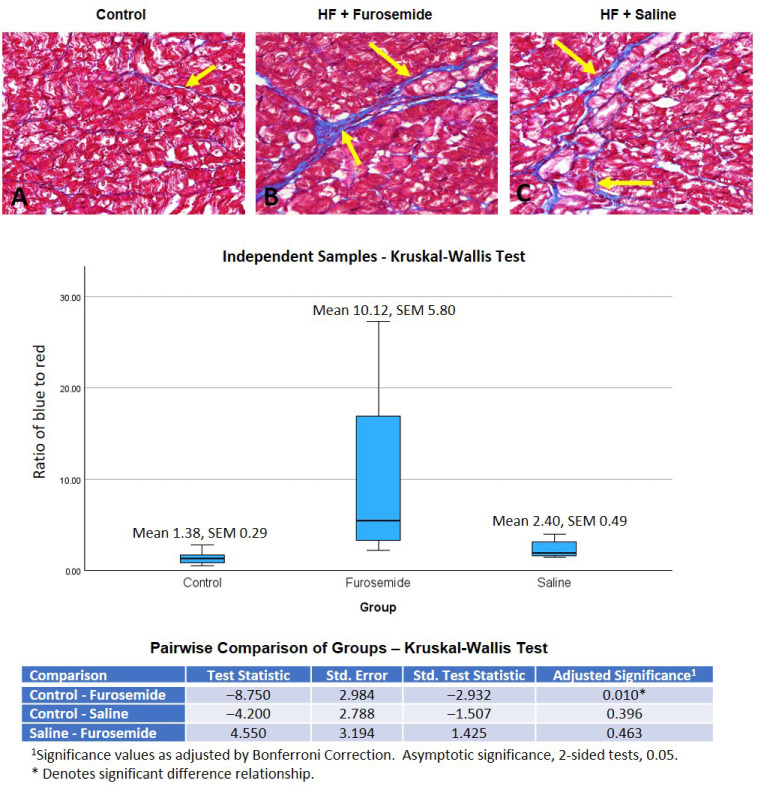
Trichrome staining (Masson’s): Quantification of the proportion of myocardial fibrosis area using Masson’s trichrome staining was performed on 5 µm thin slices from paraffin-embedded histological blocks of left ventricular tissue. Representative images from each of the groups are shown in (**A**–**C**). The yellow arrows are pointing to the blue fibrotic tissue in the ECM. The heart has three main layers of tissue running in different orientations. The amount of fibrosis varies in the different layers and locations. Independent samples Kruskal–Wallis test results, shown in the table and graph, found a significant difference between the furosemide group and the controls, but not between the control and saline groups, nor the furosemide and saline groups. Group means and standard error are reported on the graph.

**Table 1 ijms-26-06088-t001:** List of antibodies used for the protein analyses.

Antibody	Company	Catalog Number
TGF-β1	Abcam, Inc. Waltham, MA, USA	ab92486
TGF-β Receptor1	Abcam, Inc. Waltham, MA, USA	ab31013
Smad2	Abcam, Inc. Waltham, MA, USA	ab228765
Smad3	Abcam, Inc. Waltham, MA, USA	ab84177
Galectin-3	Abcam, Inc. Waltham, MA, USA	ab31707
CTGF	Abcam, Inc. Waltham, MA, USA	ab5097
PAI-1	Abcam, Inc. Waltham, MA, USA	ab66705
MMP-2	Abcam, Inc. Waltham, MA, USA	ab97779
MMP-14	Abcam, Inc. Waltham, MA, USA	ab38971
GDF-15	Abcam, Inc. Waltham, MA, USA	ab105738
β-actin	Abcam, Inc. Waltham, MA, USA	ab8224
p44/42 MAPK (ERK1/2)	Cell signaling Danvers, MA, USA	9102
TIMP-1	Sigma-Aldrich Burlington, MA, USA	IM32
P38 MAPK	ThermoFisher Scientific Waltham, MA USA	66234
MEK-1	Abcam, Inc. Waltham, MA, USA	ab109556
Ras	Abcam, Inc. Waltham, MA, USA	ab221163
JNK1/JNK2/JNK3	Abcam, Inc. Waltham, MA, USA	PA5-99528
Collagen I	Abcam, Inc. Waltham, MA, USA	ab34710
Collagen III	Abcam, Inc. Waltham, MA, USA	ab7778
Collagen VI	Abcam, Inc. Waltham, MA, USA	Ab6588

## Data Availability

The data presented in this study are available on request from the corresponding author. The data are not publicly available due to University policy.

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
