# Peer review of "Furosemide Promotes Inflammatory Activation and Myocardial Fibrosis in Swine with Tachycardia-Induced Heart Failure"

_ijms, 2025, doi:10.3390/ijms26136088_

Round 1

Reviewer 1 Report

Comments and Suggestions for Authors

This is a very elegant study on some adverse effects of furosemide examined in the pig model of tachycardia and HF model.

We know clinically that furosemide is beneficial in decongestion but can lead to activation of RAAS mainly due to inducing diuresis, thereby volume loss that in turn activates RAAS with its adverse effects such as fibrosis, LV hypertrophy etc. 

The study is generally well-written with some important implications/mechanism behind negative effects of furosemide with respect to RAAS activation.

1. Did you measure urine output and volume depletion in experimental animals and did this correlate with the level of remodeling?

2. Did you measure weight loss in these animals? Having ad libitum water feeding and no sodium restriction is a limitation in this study. Do authors have any data these animals (saline vs. furosemide infusion) had similar fluid and salt intake?

3. Did the authors consider that RAAS activation that we know activates pro-fibrotic pathways was primarily caused by relative hypovolemia (as body sees diuresis and water/salt loss as a threat to the organism and activates compensatory mechanisms)? Could authors shed some light and theoretical discussion on this framework which wasn't really considered. 

4. Would adding RAAS inhibitor to furosemide blunt adverse effects of furosemide and perhaps this combined therapy would yield net favorable results in HF model. This should be discussed as this is commonly used in clinical scenarios when treating HF patients. There is a neurohumoral blockade with these agents on top of furosemide use that we utilize for decogenstion.

5. A translational outlook should be strengthened in this manuscript and considered more in the discussion. What would this clinically imply?

Reviewer 2 Report

Comments and Suggestions for Authors

This is a very important study that examines the adverse effects of furosemide on heart failure in a pig model that may more resemble humans than other animal models.

Specific comments

In all the figures, beta-actin signals are saturated. Please show images with lighter exposures.

Fig. 1A: Although the bar graph shows significant increase in p38 MAPK expression by "HF Saline", the representative image does not show such an increase.

Figs. 4A and 4B: Although the bar graph shows that MMP-14 expression is significantly increased in both “HF Furosemide” and “HF Saline”, the representative image shows that neither increase the expression.

Fig. 6: Please indicate the results of statistical tests in the bar graph.

Materials and Methods (Animals): What do authors mean by “pigs of either gender”? Please specify the number of male animals and female animals used for each group. 

Round 2

Reviewer 2 Report

Comments and Suggestions for Authors

This reviewer thinks that the representative results should be consistent with the results in the bar graphs, but authors' points are also understandable. This would be an editorial decision.
